# Combination of VMD Mapping MFCC and LSTM: A New Acoustic Fault Diagnosis Method of Diesel Engine

**DOI:** 10.3390/s22218325

**Published:** 2022-10-30

**Authors:** Hao Yan, Huajun Bai, Xianbiao Zhan, Zhenghao Wu, Liang Wen, Xisheng Jia

**Affiliations:** Shijiazhuang Campus, Army Engineering University of PLA, Shijiazhuang 050003, China

**Keywords:** diesel engines, acoustic signals, VMD, MFCC, LSTM, fault diagnosis

## Abstract

Diesel engines have a wide range of functions in the industrial and military fields. An urgent problem to be solved is how to diagnose and identify their faults effectively and timely. In this paper, a diesel engine acoustic fault diagnosis method based on variational modal decomposition mapping Mel frequency cepstral coefficients (MFCC) and long-short-term memory network is proposed. Variational mode decomposition (VMD) is used to remove noise from the original signal and differentiate the signal into multiple modes. The sound pressure signals of different modes are mapped to the Mel filter bank in the frequency domain, and then the Mel frequency cepstral coefficients of the respective mode signals are calculated in the mapping range of frequency domain, and the optimized Mel frequency cepstral coefficients are used as the input of long and short time memory network (LSTM) which is trained and verified, and the fault diagnosis model of the diesel engine is obtained. The experimental part compares the fault diagnosis effects of different feature extraction methods, different modal decomposition methods and different classifiers, finally verifying the feasibility and effectiveness of the method proposed in this paper, and providing solutions to the problem of how to realise fault diagnosis using acoustic signals.

## 1. Introduction

Diesel engines, as the main power source in the current industrial, military and other fields, are prone to various types of failures due to their harsh working environment in many practical applications. Most of the current research on diesel engine fault identification is realized by analyzing vibration signals [1,2,3]. In practical engineering applications, diesel engines will generate high temperature or even explode during operation. Directly measuring the vibration state of the diesel engine in space is very likely to damage the sensor and transmission line, resulting in measurement distortion or even inability to measure. Acoustic measurement [4] can effectively avoid such problems and provide convenience for data analysis and collection.

At present, fault diagnosis and state identification based on acoustic signals has become an emerging research hotspot in the research of fault prediction and health management (PHM) of machinery such as diesel engines. Some experts and scholars have carried out certain research and exploration on diesel engine fault diagnosis based on acoustic signals. Mathew et al. [5] conducted combustion fault diagnosis for engine acoustic signals, established and compared three fault classification models, and weighed the diagnostic accuracy and diagnostic time of different models, which can be applied to different scenarios. Figlus et al. [6] used discrete wavelet transform to denoise the engine acoustic signal and extract entropy features, which can effectively diagnose diesel engine valve faults based on instantaneous entropy. Ning et al. [7] proposed a new dislocation superposition method, which improves the extraction accuracy of fault components by increasing the number of superpositions to the acoustic signal, and uses the similarity coefficient to automatically determine the fault components. Zhang et al. [8] proposed a parallel sparse filtering method to extract the fault features of bearings, and verified the effectiveness of the method through simulation and experiments. The above studies show that it is practical to diagnose diesel engine faults by means of acoustic signals, but in the process of these actual studies, some deficiencies and problems of non-contact data acquisition methods and traditional feature extraction methods are also exposed:(1)The sound field environment is complex, mixed with noise signals from other sound sources, resulting in a low signal-to-noise ratio in the acoustic signal, and it is difficult to separate and extract the fault information in the acoustic signal. The transmission path of sound is complex and the signal is weak, so it is difficult to locate the fault source for complex equipment.(2)Using conventional feature extraction methods to extract fault features from sound signals is still relatively one-sided and easy to ignore acoustic characteristics.

Variational mode decomposition (VMD) is widely used in signal noise reduction in the field of mechanical vibration, and some current researches also use it in acoustic signals. Facing the problem of abnormal sound recognition during vehicle operation, Kwon et al. [9] used VMD to effectively distinguish the sound into background noise and abnormal sound, and proved the good decomposition performance of VMD for sound; Chen et al. [10] proposed a method combining VMD, fast independent component analysis, and Hilbert transform to optimize the decomposition effect for the problem of diesel engine sound source decomposition, and verified the effectiveness of the method; Zhang et al. [11] optimized the decomposition performance of wind turbine aeroacoustic signals collected by a single acoustic sensor by improving the number of decomposition layers of VMD, and achieved a better blind source separation effect. These studies have shown that VMD can reduce noise, decompose effective information, and improve the signal-to-noise ratio, but it has disadvantage of not being able to directly extract the required fault features and needing to combine more specific feature extraction methods. It will lead to a substantial increase in the dimension of the features extracted later.

Mel Frequency Cepstrum Coefficient (MFCC) is a typical voiceprint recognition feature and has strong ability to extract acoustic signal features. At present, some scholars have begun to extend MFCC to the acoustic signal analysis of mechanical equipment. Suman et al. [12] proposed an algorithm combining adaptive Kalman filter and MFCC for mechanical fault detection of vehicle acoustic signals. Gong et al. [13] extracted MFCC features from the dynamic acoustic signals of vehicles, and then implemented fault feature classification using various machine learning methods. Márquez-Molina et al. [14] extracted features from the sound signal of aircraft taking off through 1/24 octave analysis and MFCC, which can effectively realize the classification of aircraft engines. It can be seen that extracting MFCC from the sound signal as a fault feature can better reflect the status information of the equipment, and it is more suitable for the characteristics of the acoustic signal, but these studies did not preprocess the original acoustic signal, which will lead to noise during the feature extraction process. Greater impact.

The combined use of VMD and MFCC can not only improve the signal-to-noise ratio of the signal, but also extract fault features that conform to the acoustic characteristics, but it will lead to a high feature dimension and introduce some useless features, which will introduce interference for later diagnosis and identification. and increase the computational burden. Therefore, this paper proposes a diesel engine fault feature extraction method based on VMD mapping MFCC. This method uses VMD to decompose the mode of the original one-dimensional sound pressure signal, removes useless noise components according to the change of the energy ratio of each intrinsic mode function (IMF), and maps the reserved IMFs to the frequency distribution of the Mel filter bank in the frequency domain. The Mel filter corresponding to each IMF is used to calculate the MFCC as the fault feature of the diesel engine, which effectively solves the problem of fault information extraction. In view of the memory function of the LSTM network, it has a strong learning ability in time series data [15,16]. Therefore, the double-layer LSTM network is selected as the fault diagnosis classifier, which effectively solves the problem of fault classification and localization. Combined with the diesel engine preset fault experiment, compared with the traditional feature extraction method and the unimproved MFCC extraction method, the method proposed in this paper can effectively characterize the diesel engine state and reduce the feature dimension. Then the diagnostic effects of different classification networks are compared to test the effectiveness, superiority and robustness of the proposed model.

The main contributions and innovations of this paper are as follows:(1)A feature extraction method of VMD mapping MFCC is proposed. According to the corresponding relationship between the IMF component and the frequency distribution of the Mel filter bank, the corresponding MFCC is calculated to form a fault feature. The feature has a low dimension and a strong ability to represent the state, which can effectively improve diagnostic accuracy.(2)The deep learning method is introduced to train the classification model, and the double-layer LSTM network is selected as the output classifier for network training and parameter fine-tuning, which effectively reduces the network training time and improves the accuracy of fault identification.(3)Taking the acoustic signal as the research object, combined with VMD mapping MFCC and LSTM network to build a diesel engine fault diagnosis model, the model shows more efficient and accurate diagnosis effect in the experiment.

The main contents of the other sections of this paper are as follows: Section 2 introduces the basic theory of VMD, MFCC and LSTM in detail; Section 3 describes the implementation process of the fault diagnosis model based on VMD mapping MFCC and LSTM networks; Section 4 describes the setup and result analysis of the diesel engine preset fault test; the conclusion is showed in Section 5.

## 2. Basic Theory

### 2.1. Variational Mode Decomposition

VMD is a signal mode decomposition method proposed by Zosso’s team in 2014 [17]. It has the characteristics of self-adaptation and non-recursion. As a new non-stationary, nonlinear signal processing method, replacing the previous local mean decomposition (LMD) [18], empirical mode decomposition (EMD) [19], ensemble empirical mode decomposition (EEMD) [20] and other recursive decomposition modes, which improve the mode mixing and end-effect problems of traditional decomposition methods by solving constrained variational problems. Literature [21,22] verified that VMD has better complex data decomposition accuracy and better anti-noise interference, etc.

The function of the VMD algorithm is to construct and solve the variational problem, decompose the original signal *f*(*t*) into *K* IMF components $u_{k}(t)$. Under the condition of the sum of each component is equal to the input signal, the center frequency and bandwidth are continuously updated through the iterative process, and finally the IMF component that minimizes the sum of the IMF bandwidths is obtained. The specific steps are as follows:

(1) Through the Hilbert transform, the one-sided spectrum of the analytical signal of each IMF component $u_{k}(t)\;(k=1,2,\cdot \cdot \cdot ,K)$ is obtained:(1)$$\left[\delta (t)+(j/\pi t)]*u_{k}(t\right)$$

Adjust the position of the center frequency of each intrinsic mode to the respective baseband:(2)$$\left\{\left[\delta (t)+(j/\pi t)]*u_{k}(t\right)\right\}e^{-j\omega_{k}t}$$
where δ(t) is the impulse function, ∗ is the convolution calculation, $u_{k}(t)$ is the *k*th IMF component and $\omega_{k}$ is the center frequency of each IMF component.

(2) The L2-norm of the above demodulated signal gradient is calculated, and the bandwidth of each IMF component is estimated. Get the constrained variational model expression:(3)$$\left\{\begin{array}{c} \underset{\left\{u_{k},\omega_{k}\right\}}{\min}\left\{\sum_{k=1}^{K}{\left\|\partial_{t}\left[\left(\delta (t)+\frac{j}{\pi t})*u_{k}(t\right)\right]e^{-j\omega_{k}t}\right\|}_{2}^{2}\right\}\\ s.t.\sum_{k=1}^{K}u_{k}=f(t) \end{array}\right\}$$

(3) The quadratic penalty factor α and the Lagrange multiplication operator λ are introduced into the variational solution problem to transform the constrained variational problem into an unconstrained variational problem. The augmented Lagrange function is as follows:(4)$$\begin{array}{r} L(\{u_{k}\},\{\omega_{k}\},\lambda )=\alpha {\sum_{k=1}^{K}\left\|\partial_{t}\left[\left(\delta (t)+\frac{j}{\pi t})*u_{k}(t\right)\right]e^{-jw_{k}t}\right\|}_{2}^{2}\\ +{\left\|f(t)-\sum_{k=1}^{K}u_{k}(t)\right\|}_{2}^{2}+\langle \lambda (t),f(t)-\sum_{k=1}^{K}u_{k}(t)\rangle \end{array}$$
where α is usually selected to be a large enough positive number to improve the reconstruction accuracy of the signal; λ(t) ensures the strict restriction of the constraints; ${\left\|\right\|}_{2}^{2}$ represents the operation of L^2^-norm and 〈 〉 represents the operation of inner product.

Alternate direction method of multipliers (ADMM) is used to update the value of each $u_{k}$,$\omega_{k}$.and calculate the saddle point of the augmented Lagrange function. It is the way to find the optimal solution of the constrained variational model, so as to realize modal decomposition. The iterative process of $u_{k}$,$\omega_{k}$ is as follows:(5)$${\hat{u}}_{k}^{n+1}\left(\omega \right)=\frac{\hat{f}\left(\omega \right)-\sum_{i\neq k}{\hat{u}}_{i}^{n}\left(\omega \right)+\frac{\hat{\lambda }\left(\omega \right)}{2}}{1+2\alpha {\left(\omega -\omega_{k}\right)}^{2}}$$
(6)$$\omega_{k}^{n+1}=\frac{\int_{0}^{\infty }\omega {\left|{\hat{u}}_{k}^{n+1}\left(\omega \right)\right|}^{2}d\omega }{\int_{0}^{\infty }{\left|{\hat{u}}_{k}^{n+1}\left(\omega \right)\right|}^{2}d\omega }$$
where ${\hat{u}}_{k}$ is the IMF function in the frequency domain state, $\hat{\lambda }$ is the Lagrange multiplication operator in the frequency domain state.

In the VMD process, the difference of the number of decomposition layers *K* will affect the effect of modal decomposition, which is a limitation of VMD. The determination of the number of decomposition layers *K* needs to be set manually, which has a certain degree of randomness and subjectivity. Therefore, this paper will determine the appropriate *K* value by calculating the spectral centroid of each IMF.
(7)$$\omega_{G}^{k}=\frac{\int_{0}^{\infty }\omega \left|{\hat{u}}_{k}\left(\omega \right)\right|d\omega }{\int_{0}^{\infty }\left|{\hat{u}}_{k}\left(\omega \right)\right|d\omega }$$
where $\omega_{G}^{k}$ is the spectral centroid of each *k*th IMF.

### 2.2. Mel Frequency Cepstrum Coefficient

MFCC was first proposed by Davis and Mermelstein in the 1980s [23], and this study effectively proved that the coefficient has a better identification effect than other parameters. So far, MFCC has been widely used in various fields of speech recognition, including instruction recognition, emotion recognition, and person recognition. At the same time, MFCC is also gradually introduced into the status and fault identification of some mechanical equipment, such as transformers [24], UAVs [25], etc., and good results have also been achieved by means of MFCC.

The physical meaning of MFCC, in simple terms, is a cepstral coefficient calculated in the Mel-scale frequency domain, which can be understood as the parameter corresponding to the energy of the sound signal in different frequency ranges in the cepstrum, which can reflect low frequency envelope and high frequency detail information. The Mel-scale frequency domain is a frequency domain that simulates the human ear’s perception of sound frequency. Its definition formula is as follows:(8)$$M\left(f\right)=2595\mathrm{lg}\left(1+\frac{f}{700}\right)$$
where *M* is the Mel frequency; *f* is the frequency.

The MFCC extraction process of the sound signal is shown in Figure 1, in which the pre-emphasis is to compensate the high-frequency components of the signal and prevent the high-frequency components from being attenuated or even lost in the subsequent calculation process. When framing the sound pressure signal, considering the continuity between the two frames, there should be an overlapping area between the two adjacent frames. In speech recognition, the frame length is generally set to 25 ms, and the frame shift is set to 10 ms [26]. The acoustic signal of the diesel engine is relatively more stable, so this paper also uses this setting. The framed signal is windowed with a Hamming window to enhance the continuity of the signal and reduce the distortion caused by the subsequent FFT. The Hamming window formula is expressed as follows.
(9)$$W\left(a\right)=\left\{\begin{array}{ll} 0.54-0.46\cos\left(\frac{2\pi a}{A-1}\right), & 0\leq a\leq A-1\\ 0, & a<0\&a>A-1 \end{array}\right.$$
where *A* is the length of the Hamming window.

The Mel filter is composed of several triangular bandpass filters. As shown in Figure 2, if the number of filters is *N*, then it is necessary to determine the *N* + 2 Mel frequency domain scale values *M*(0), *M*(1), *M*(2),…, *M*(*N*), *M*(*N* + 1), which are equally spaced values. Through the inverse operation of Equation (8), the scale value in the linear frequency domain is obtained: *f*(0), *f*(1), *f*(2),…, *f*(*N*), *f*(*N* + 1). According to the *N* + 2 scale values, the functional expression of the *N* Mel filters can be calculated
(10)$$H_{n}\left(k\right)=\left\{\begin{array}{ll} 0 & k<f\left(n-1\right)\\ \frac{2\left(k-f\left(n-1\right)\right)}{\left(f\left(n+1\right)-f\left(n-1\right)\right)\left(f\left(n\right)-f\left(n-1\right)\right)} & f\left(n-1\right)\leq k\leq f\left(n\right)\\ \frac{2\left(k-f\left(n-1\right)\right)}{\left(f\left(n+1\right)-f\left(n-1\right)\right)\left(f\left(n+1\right)-f\left(n\right)\right)} & f\left(n\right)\leq k\leq f\left(n+1\right)\\ 0 & k\geq f\left(n-1\right) \end{array}\right.$$
where $H_{n}\left(k\right)$ is the function of the *n*th Mel filter, and *k* is the independent variable of the function, that is, the frequency.

After the preprocessed signal is filtered by the Mel filter bank, *N* parameters $m_{n}$ (*n* = 1, 2,…, *N*) can be obtained:(11)$$m_{n}=\sum_{k=0}^{F-1}\left|X(k)\right|\times H_{n}(k),\;n=1,2,\ldots ,N$$
where *F* is the number of FFT points; X(k) is the frequency domain function of the preprocessing framed signal after FFT. Take the logarithm of $m_{n}$ calculated by Equation (11), and perform DCT on it. The transformation process is as follows:(12)$$c(n)=\sqrt{\frac{2}{F}}\sum_{j=1}^{N}\ln m_{j}\cos[\frac{\pi n(2j-1)}{2N}],\;1\leq n,j\leq N$$
where c(n) is the MFCC of each framed signal.

However, the traditional MFCC extraction method realizes frequency domain conversion through FFT. In this process, it is assumed that the signal is approximately unchanged in a short time, and the nonlinearity and non-stationarity of the signal cannot be reflected, resulting in the loss of some information. Decomposing the signal by VMD and then extracting the MFCC will reflect the local characteristics more accurately, so as to obtain more comprehensive diesel engine fault characteristics.

### 2.3. VMD Mapping MFCC Features

Due to the low signal-to-noise ratio of the diesel engine sound signal, MFCC, as features, can effectively reflect the envelope and detail information of the diesel engine acoustic signal. but the conventional MFCC feature extracted from the original signal directly without noise reduction may result in more interference in the features. Therefore, multiple IMF components are obtained after VMD of the original signal, and the noise components are removed from them, which can achieve effective noise reduction of the original signal. Zou et al. [27] extracted MFCC and GFCC from each IMF component decomposed by the VMD of the UAV noise signal, and formed a mixed feature together with the energy ratio of each IMF component of the VMD. Drone audio can be identified effectively in different noise environments. However, this method of directly extracting features from the original signal may still contain a large amount of noise components in the features. At the same time, the method of extracting MFCC features for all IMF components will lead to higher dimensions of the extracted features and increase the computational burden. And even some cepstral coefficient features that are not conducive to classification are introduced. Considering that the distribution of each IMF component is different in the frequency domain, it is usually concentrated in the frequency range of 4 to 5 triangular Mel filters, as shown in Figure 3. The MFCC of each frame signal is also calculated separately by each triangular Mel filter.

If only the cepstral coefficients corresponding to the main frequency distribution filters of each IMF component are calculated during the MFCC calculation process of the IMF components, some invalid information can be eliminated, the feature dimension can be reduced, and the operation speed can be improved. Therefore, this paper proposes a diesel engine fault feature extraction method based on VMD Mapping MFCC (VMMFCC). The extraction process of VMMFCC is as follows:

The IMF component is FFT, and its frequency domain distribution is mapped to the frequency distribution of the Mel filter bank, so that each IMF_i1_ mainly corresponds to 2~4 Mel triangular filters in frequency, for example, the first IMF_i1_ reserved corresponds to the *x*_1_th to *y*_1_th filters. Likewise, the *n*th IMF*_in_* retained corresponds to the *x_n_*th to *y_n_*th filters. Use the triangular filters corresponding to the IMF components to calculate the MFCC of the IMF, and combine them to form a new cepstral coefficient feature: [*c*(*x*_1_), *c*(*x*_1_ + 1),…, *c*(*y*1), *c*(*x*_2_), *c*(*x*_2_ + 1),…, *c*(*y*_2_),…, *c*(*x_n_*), *c*(*x_n_* + 1),…, *c*(*y_n_*)], namely VMMFCC.

### 2.4. Long Short-Term Memory Network

Recurrent neural network (RNN) is the most traditional method of deep learning for time series signal classification, but RNN often has the problem of gradient disappearance for long sequence data, so it is difficult to learn and save long-term information. LSTM [28] proposed by Hochreiter and Schmidhuber in 1997 solves this problem well, and its core is a special implicit neural unit, as shown in Figure 4. LSTM can effectively prevent the occurrence of gradient disappearance and gradient explosion, and is widely used in long sequence recognition. It means that LSTM is also applicable to acoustic signals belonging to long time series.

Compared with RNN, LSTM is more complex. On the basis of retaining the original hidden state *h_t_*, it also adds a new hidden state of the cell state, denoted as *C_t_*. The long-term memory and short-term memory functions of LSTM are realized through the gated structure, including input gate, forget gate and output gate. The specific calculation process is as follows:

(1) The forget gate is a gated structure that controls the cellular state at the previous moment to be forgotten according to the probability. The gate will read the hidden state of the previous moment and the input of the current moment, and output a value between 0 and 1 to represent the probability that the data is retained, the formula is as follows:(13)$$f_{t}=\sigma (W_{f}\cdot [h_{t-1},x_{t}]+b_{f}),\;f_{t}\in [0,1]$$
where *h_t_*_−1_ represents the hidden state at the previous moment, *x_t_* is the input at the current moment, W is the weight vector, b is the bias vector, σ represents the sigmoid function, and *f_t_* is the retention probability of the cell state.

(2) The input gate is a gated structure that determines the degree to which new information enters the cellular state. The gate consists of two parts. The calculation formulas are as follows:(14)$$i_{t}=\sigma (W_{i}\cdot [h_{t-1},x_{t}]+b_{i})\;$$
(15)$${\tilde{C}}_{t}=\tanh(W_{C}\cdot [h_{t-1},x_{t}]+b_{C})\;$$

Equation (14) uses the sigmoid layer to decide which information needs to be updated, that is, the update probability: *i_t_*; Equation (15) uses the tanh layer to generate the update content of the current moment: ${\tilde{C}}_{t}$. The results of the two equations are multiplied to update the cellular state.

(3) Combine the forget gate and the input gate to calculate the current cell state *C_t_*.
(16)$$C_{t}=C_{t-1}⊙f_{t}+i_{t}⊙{\tilde{C}}_{t}$$

(4) The output gate is a gated structure that determines the final output of the cell state to the hidden state *h_t_*. It is also composed of two parts. The calculation is as follows:(17)$$o_{t}=\sigma (W_{o}\cdot [h_{t-1},x_{t}]+b_{o})\;$$
(18)$$h_{t}=o_{t}⊙\tanh(C_{t})$$

Equation (17) uses the sigmoid layer to determine which parts of *C_t_*. will be output. Equation (18) is to multiply the output of the sigmoid layer with the *C_t_*. processed by tanh to obtain the hidden state output *h_t_* at the current moment.

## 3. Fault Diagnosis Process Based on VMMFCC and LSTM

When calculating the MFCC for the IMF components, only the cepstral coefficients of the Mel filter corresponding to the frequency distribution of each IMF component are calculated. It can eliminate some invalid information, reduce the feature dimension, and improve the operation speed. Therefore, this paper proposes a diesel engine fault diagnosis method based on VMMFCC-LSTM. The flow chart is shown in Figure 5. The specific implementation steps are as follows:

**Step 1:** Optimization of VMD decomposition layers. Under different decomposition layers K, perform VMD on the original sound pressure signal of the diesel engine, and calculate the spectral centroid of each IMF component. When the boundary of the spectral centroid tends to be stable for the first time, VMD can not only avoid under-decomposition, but also better avoid modal aliasing. Therefore, the K value at this time is determined as the number of decomposition layers of VMD.

**Step 2**: IMF component optimized selection. the capacity ratio and energy variation range of each IMF under different operating states of the diesel engine are calculated to determine which IMF components contain large amounts of information and are more sensitive to fault conditions, and retain these n IMF components for subsequent feature extraction.

**Step 3**: Map the IMF frequency distribution to the Mel filter bank and calculate the new feature vectors. FFT is performed on the reserved IMF components, and the distribution in the frequency domain is mapped to the frequency distribution of the Mel filter bank, so as to calculate VMMFCC.

**Step 4**: Feature set partitioning. The feature sets of the diesel engine sound pressure signals in different operating states are extracted, and the training set and the validation set are divided in each state.

**Step 5**: Train the LSTM classification network. Build the LSTM network and adjust the network parameters, input the training set, and train the LSTM network classifier.

**Step 6**: Fault identification. The trained LSTM network structure and parameters are transferred to the classification network of the validation set to verify the effect of the proposed method.

**Figure 5 sensors-22-08325-f005:**
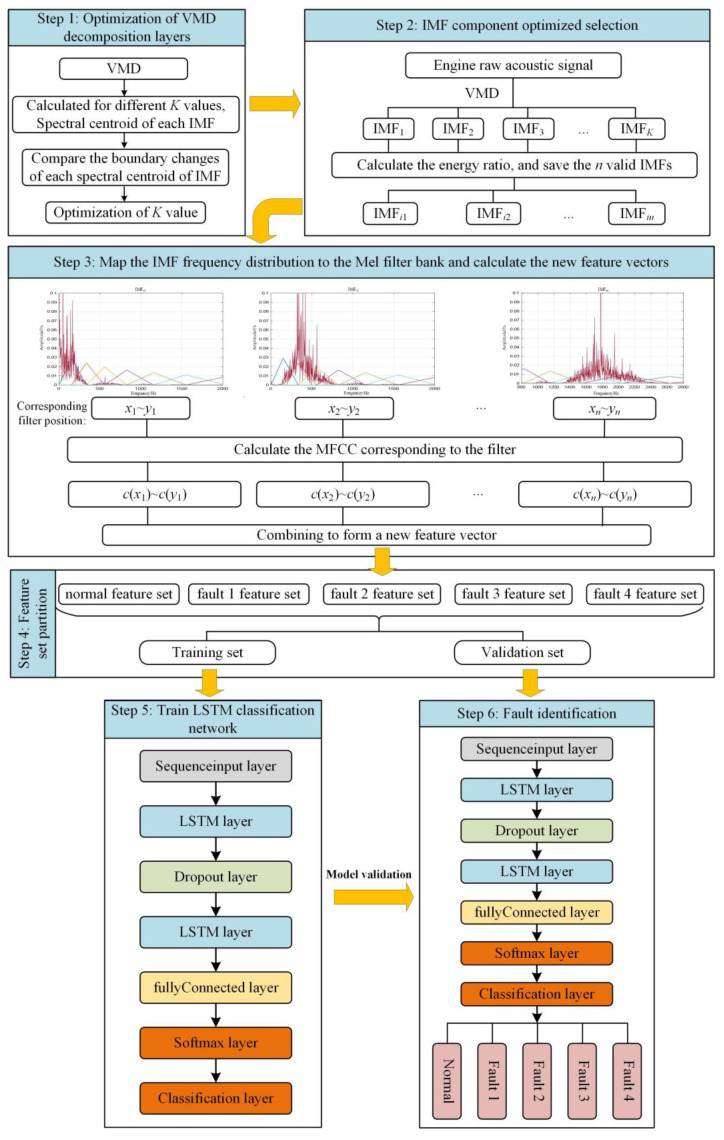
Full flowchart of fault diagnosis.

## 4. Diesel Engine Preset Experiment

### 4.1. Experimental Setup

The diesel engine preset experiment takes a high pressure common rail diesel engine with the model of CA6DF3-20E3 as the research object. This test bench is a customized derating test bench for experiment and teaching. In the artificial preset fault experiment, the sound pressure signals of the diesel engine under five working conditions, namely, normal, blocked air inlet, the first cylinder misfire, the third cylinder misfire and the sixth cylinder misfire, are tested respectively. The installation position of the test bench and the sensor is shown in Figure 6.

According to the standard on the position of the microphone in GB/T 17248.3-2018, the distance between the two sensors does not exceed 2 m and the distance between the sensor and the ground shall not be less than 1.2 m. Therefore, the two sound pressure sensors are arranged at the center of 30 cm on each side of the diesel engine, and the height is basically the same as the upper plane of the cylinder head. The experiment adopts YSV5001 high-precision ICP-type sound pressure sensor and DH5902N data acquisition system. The sensor sensitivity is 50 mV/Pa and the sampling frequency is 20 kHz. In the abnormal state setting, the blockage of the intake port is realized by installing the intake cover, and the misfire is realized by disconnecting the ignition power cord of the corresponding cylinder. According to the sampling settings in Table 1, the sound pressure signals of the diesel engine running in five states were collected.

### 4.2. Experimental Data Preprocessing

The sound pressure signals of different states in the time domain are shown in Figure 7a, and it is found that the periodicity and regularity of the sound pressure signals of diesel engines are poor. Through the fast Fourier transform, the frequency domain signal diagram is obtained as shown in Figure 7b. It can be seen that the spectral distribution difference under different states is not too large, so it is difficult to find the corresponding fault characteristic frequency. It can be found that the spectral energy is mainly distributed within 9 kHz, which does not exceed the hearing range of the human ear of 20Hz~20 kHz, so it basically conforms to the applicable range of the Mel filter bank.

#### 4.2.1. Data Set Partitioning

In order to facilitate the analysis and verification of the scientificity and validity of the classification model constructed in this paper, it is necessary to divide the collected data. In this experiment, the 0.5 s sound pressure data collected by the sound pressure sensor 1 is used as a sample, that is, 10,000 sampling points are used as a sample. Therefore, the number of samples measured in each state is 120, of which 100 samples are selected as the training set, and the remaining 20 samples are used as the verification set. The data set division is shown in Table 2.

#### 4.2.2. The Extraction of VMMFCC

Compared with the traditional EMD, the VMD can effectively improve the modal aliasing phenomenon, but the determination of the decomposition level *K* of the VMD needs to be set manually. For complex signals, the value of *K* will directly affect the decomposition effect. If its value is too large, it will cause over-decomposition and increase the degree of modal aliasing; if its value is too small, it will lead to under-decomposition and no useful signal will be obtained. In this paper, the optimal number of decomposition layers is determined by analyzing the distribution of the spectral centroid of each IMF component. The formula for calculating the spectral centroid is as follows:(19)$$P=\frac{\sum_{k=0}^{N-1}k\left|S(k)\right|}{\sum_{k=0}^{N-1}\left|S(k)\right|}$$
where *N* is the number of FFT points, *k* is the frequency subscript, and *S*(*k*) is the amplitude value of the frequency domain signal at *k*.

VMD is performed on the sound pressure signal according to different decomposition layers *K*, and the respective spectral centroids of the IMFs are calculated respectively, and they are arranged in order from low to high. The results of normal status are shown in Table 3. As the value of *K* increases, the range boundary of the spectral centroid will gradually stabilize. When the boundary values of the spectral centroid are stable for the first time, it can be determined that the *K* value at this time is the optimal number of decomposition layers. When *K* = 8, the boundary value of the spectral centroid begins to stabilize. Therefore, for the collected sound pressure signal, the optimal value of *K* is determined to be 8. At this time, the decomposition effect of VMD is shown in Figure 8. The spectral distribution of each IMF has no obvious modal aliasing and is relatively concentrated, so the decomposition effect is ideal.

In order to verify whether the optimal decomposition level K = 8 is also applicable to other data of fault state, acoustic signal is processed by VMD under different states with different K-values, according to the same method, and calculate the spectral centroids of each IMF respectively. Since the change of boundary value of spectrum centroid is mainly considered when determining the optimal number of layers, the maximum and minimum values of the IMF spectrum centroid of each group of signals are calculated, as shown in Table 4. By observing the same state data, with the constant increase of *K*, the maximum value of spectral centroid tends to be stable after *K* = 8, and when *K* = 8, the minimum value of spectral centroid is also in the stable range. Therefore, *K* = 8 is the optimal decomposition level for VMD processing of diesel engine operation data in different states.

VMD of the original sound pressure signal only decomposes the signal into multiple modes, but it cannot play a role in noise reduction. Therefore, it is also necessary to filter and eliminate the IMF components obtained, so as to achieve signal noise reduction and retain the useful part of the signal. The noise mentioned here is generalized. Any component that is not conducive to fault feature extraction can be regarded as noise. When the energy proportion of an IMF component changes significantly under different operating conditions, it can be considered that the IMF component is more sensitive to fault conditions. At the same time, it is also necessary to take into account the amount of information contained in the IMF. If the energy of the IMF component is relatively small, it means that its amount of information is small. Therefore, IMFs with a large proportion of energy and obvious changes among different states should be retained.

In this paper, Average proportion of energy (APE), variance of contribution of energy (VPE) and average Pearson correlation coefficient (APCC) are selected as the selection indicators of IMF components, in which APE is obtained by averaging the energy proportion of the same IMF in all groups of data; The VPE is obtained by first calculating average energy proportion of the same IMF in 120 groups of data under each state, and then calculating the variance between the five average values. Therefore, VPE can reflect the sensitivity of the IMF to changes in operating conditions to a certain extent; The APCC is obtained by averaging the Pearson correlation coefficients of the same IMF and the original signal in each group of data, which can reflect the correlation between IMF components and the original signal.

Table 5 shows APE, VPE and APCC of IMF_1_~IMF_8_. In this paper, five IMF components that are larger in each index value are selected as candidates. Among them, the APE of IMF_1_~IMF_5_ is relatively high, indicating that these five components contain relatively large amount of information; The VPE of IMF_1_, IMF_2_, IMF_4_, IMF_5_ and IMF_6_ is large, which means that these five components are more sensitive to fault conditions; The APCC of IMF_1_~IMF_5_ is high, indicating that the IMF component contains high original information content. After comprehensive consideration, four IMF components, IMF_1_, IMF_2_, IMF_4_ and IMF_5_, which take into account the three indicators, are retained in this paper to further extract fault features.

The spectra of the four reserved IMF components will be corresponding to the constructed Mel filter bank, as shown in Figure 9. The frequency distribution of each IMF component of the energy will be mainly concentrated in some filters. Take IMF_1_ as an example, its spectrum is mainly distributed in the range of the first to fourth triangular filters. Therefore, the first to fourth triangular filters are called the mapped part. And the distribution in the 5th to 13th filters is very small, which is called the unmapped part. The IMF_1_ component itself has very little information in the unmapped part, and filtering through the filter will only result in less obvious features, or even cause interference. So only the 4 MFCC values of the mapped part are calculated for IMF_1_. In the same way, the mapping relationship between different IMF components and Mel filters is obtained, as shown in Table 6.

This experiment takes 0.5 s of data as a sample, that is, 10,000 sampling points as a sample. Generally speaking, the frame length is set to 25 ms, and the frame shift is set to 15 ms. This article uses this setting to calculate the frame length as 20,000 × 0.025 = 500 sampling points, and the frame shift as 20,000 × 0.015 = 300 sampling points. Therefore, a sample will be divided into 32 frames in feature extraction, and the sum of the number of MFCCs extracted in each frame is the dimension of the frame’s feature vector, so that a set of 32 × 17 features vector is extracted from each set of data.

### 4.3. Comparison of Feature Effect

The VMMFCC feature is different from the traditional MFCC feature. Figure 10a–e shows the MFCC feature vector map in five states. From the shape of the graph, under the same feature dimension, the feature coefficients corresponding to different frames fluctuate greatly under the same feature dimension, indicating that the feature fluctuates greatly over time, so the robustness of the feature is poor. Therefore, the robustness of this feature is poor. When the MFCC is directly extracted from the original signal, the feature will also be extracted from the noise component. Therefore, the feature vectors are more cluttered and less regular, which will also bring a burden to the subsequent training of the LSTM network.

The VMMFCC feature vectors graph under five states are showed in Figure 11a–e. Under the same feature dimension, the feature coefficients of different frames are relatively stable. It can be seen that the feature coefficients at this time are less affected by noise and have stability in the time domain. Comparing Figure 11a–e, the shape and trend of VMMFCC feature vector graph is more distinct between different states, so the characteristic coefficient is more conducive to the realization of fault diagnosis in principle.

In order to verify the effectiveness of VMD for MFCC coefficient mapping selection and outperform other modal decomposition methods. Use LMD, EMD and EEMD to decompose the original signals, and then get the mapping MFCC under different mode decomposition methods through the process in Figure 5, namely LMMFCC, EMMFCC and EEMMFCC. The MFCC of the original signal and the four mapping MFCCs are trained and verified using LSTM network. The final fault diagnosis accuracy is shown in Table 7, and the fault diagnosis confusion matrix is shown in Figure 12.

It can be seen that the accuracy of fault diagnosis can reach 87% when taking MFCC directly extracted from the original signal as the feature, but the diagnosis ability for S3 state is poor. Although feature LMMCC and EMMFCC use modal decomposition and mapping, due to the limited decomposition ability of LMD and EMD, it may lead to modal aliasing, which will bring more interference to feature extraction later, so the diagnostic effect is not as good as that of MFCC. With EEMMFCC as the feature input, the diagnostic accuracy has been improved to a certain extent, reaching 90%, but each state cannot be completely accurately identified, and the feature extraction time is long. VMMFCC is the feature proposed in this paper. Its feature extraction time is slightly higher than that of other features with lower accuracy, but the overall fault diagnosis accuracy can reach 97%, and individual confusion only exists among the three similar fault states of S3, S4 and S5. Therefore, VMMFCC can better reflect the running state of diesel engine and obtain higher accuracy in the diagnosis process.

### 4.4. Comparison of Classifier Effects

After feature extraction, proper fault classification diagnostic device is also the key to fault diagnosis. For one-dimensional time-domain signals, LSTM is the most commonly used neural network recognition classification signal. The construction of LSTM network should take into account such structural factors as data format, number of layers, number of hidden units, and set appropriate learning rate, batch size, discard rate, maximum number of iterations and other network learning super parameters [29]. Since the original signal has been feature extracted in the early stage, there is no need to use the overly complex deep network for feature learning, so the simple double-layer LSTM network structure is used in this paper. Specific parameters of network model training are shown in Table 8. Other learning super parameter settings include: the decay period of learning rate decay period is 30, the decay factor of learning rate is 0.1, the batch size is 32, and the maximum number of iterations is 35.

To verify the rationality of parameter settings of LSTM network, we also compared the impact of different number of hidden cells and dropout rates on network training and final diagnosis results, as shown in Table 9. As the number of hidden layer units increases, the number of network training parameters increases. At the same time, by comparing the diagnostic accuracy of different combinations of number of hidden cells of LSTM and dropout rate, it can be seen that number of hidden cells = 100 and dropout rate = 0.2 are the best combination of parameter settings.

The t-SNE visualization process is performed on the features processed by different layers of the double-layer LSTM network, and the intuitive classification effect is shown in Figure 13. It can be seen that the distribution of fault characteristics of each state extracted by LSTM1 is generally scattered, only the S1 state can be basically distinguished, and the features of the other four states are seriously scattered, making it difficult to achieve fault separation. The role of the Dropout layer is to prevent overfitting, so the extracted features can not improve the classification effect. After the feature vector is processed by LSTM2, five states can be effectively identified. Among them, the S1 and S2 states can be completely distinguished, and there is a very small amount of confusion between the S3, S4, and S5 states, but these three states can also be effectively identified. The classification effect of the fullyConnected layer is more obvious, but there is still a very small amount of confusion between S3, S4, and S5. Considering that S3, S4, and S5 are all in the misfire state of a certain cylinder in the diesel engine, their fault states are relatively close, so there may be misjudgments in the identification process.

In order to verify the effectiveness of LSTM network model for sound pressure signals and extracted features, common classification models used in machine learning and deep learning are selected for training. The accuracy of fault diagnosis obtained by inputting verification sets into different classification models is shown in Table 10. Among them, as a traditional machine learning model, SVM has a relatively simple structure, so the sum of trainable parameters is less, but it is difficult to effectively distinguish the types of fault states; For LSTM, RNN and 1D-CNN neural networks, the total training parameters of LSTM network classifier are relatively small and can achieve better classification effect, which proves that LSTM network is more suitable for the classification of sound signals under different operating conditions of diesel engines.

The diagnosis effect of VMMFCC in different classification models is showed in Figure 14. When using SVM for diagnosis, various states cannot achieve a good fault identification effect. When using the traditional RNN model for classification, the accuracy rate can reach 91%, but the effect is not ideal. It may be that the network cannot be accurately classified due to the disappearance of the gradient during the training process. The diagnostic accuracy rate of 1D-CNN is 86%. Although the convolutional network has strong self-learning ability, it is difficult to learn the features of time series signals with strong correlation. When using LSTM, the overall accuracy rate can reach 97%, which is better than other classifiers and can achieve relatively satisfactory results.

To sum up, the VMMFCC-LSTM fault diagnosis method proposed in this paper has more representative feature extraction and better diagnosis effect, and provides a more effective method and idea for diesel engine acoustic fault diagnosis.

## 5. Conclusions

In this paper, a fault diagnosis method for diesel engine acoustic signals based on VMD mapping MFCC and LSTM is proposed. A feature that can better represent the current operating state is extracted from the sound pressure signal obtained by non-contact measurement, and then the LSTM network is trained for diesel engine fault diagnosis. The main contributions of this paper are as follows:(1)VMD on the diesel engine sound pressure signal is performed, and calculate the spectral centroid of each IMF component under different decomposition layers *K* is calculated. According to the change of the boundary of spectral centroid, the optimal number of decomposition layers is determined to realize the optimization of VMD.(2)The MFCC method is introduced into the analysis of diesel engine sound signals. According to the corresponding relationship between the frequency domain distribution of the IMF component and the Mel filter, a feature extraction method of VMD mapping MFCC is proposed for the first time, which effectively improve the ability of the sound signal of the diesel engine to reflect the fault state.(3)A diagnosis method based on two-layer LSTM network is proposed. VMMFCC is input into LSTM network for parameter pre-training, then the verification set is used to verify the LSTM network. Thus, a fault diagnosis classifier suitable for VMMFCC is constructed, which achieves good fault diagnosis effect.

The experimental results show that the proposed method is superior to other diagnostic models in diesel engine acoustic fault diagnosis. In the engineering application of pursuing high timeliness, this research can quickly judge the fault type and locate the fault location by non-contact measurement means. In addition, the results of research will help to expand new knowledge in this field, with good reference value, and provide a new idea for relevant scholars.

## Figures and Tables

**Figure 1 sensors-22-08325-f001:**
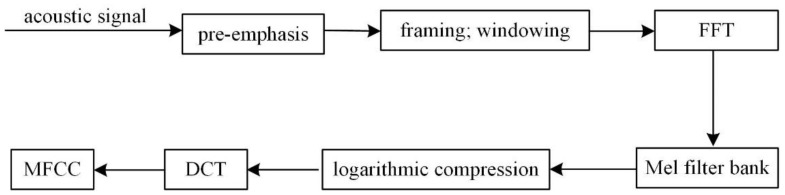
MFCC extraction process.

**Figure 2 sensors-22-08325-f002:**
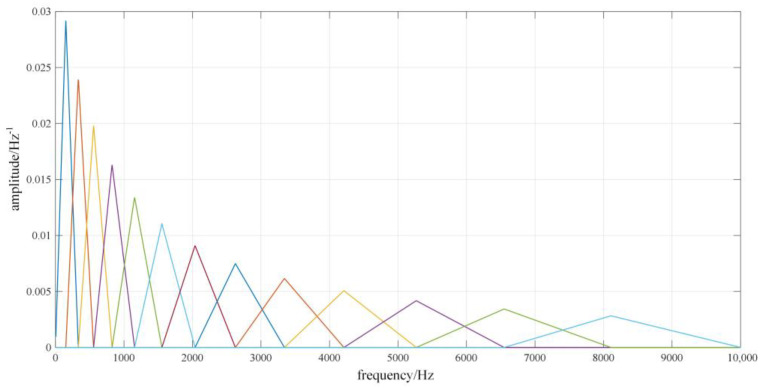
Non-equal-height Mel filter bank diagram.

**Figure 3 sensors-22-08325-f003:**
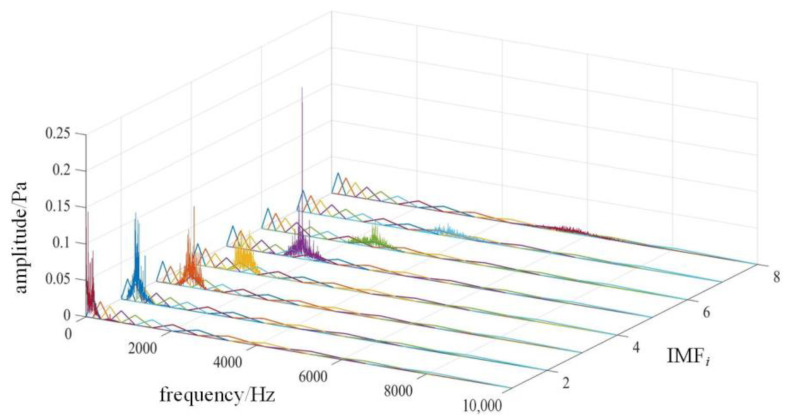
The frequency domain distribution of each IMF component is mapped on the Mel filter bank.

**Figure 4 sensors-22-08325-f004:**
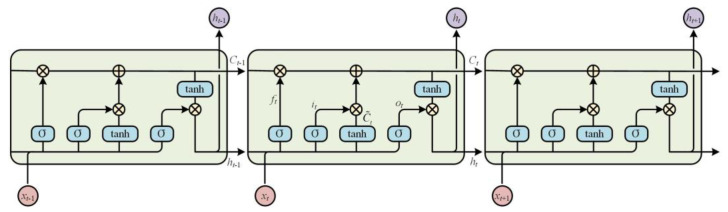
Schematic diagram of LSTM structure.

**Figure 6 sensors-22-08325-f006:**
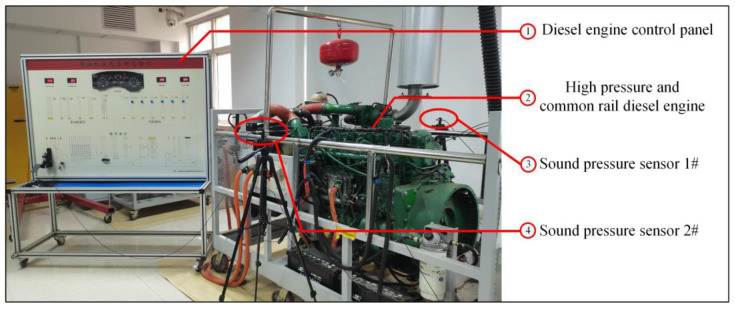
Diesel engine condition monitoring test bench.

**Figure 7 sensors-22-08325-f007:**
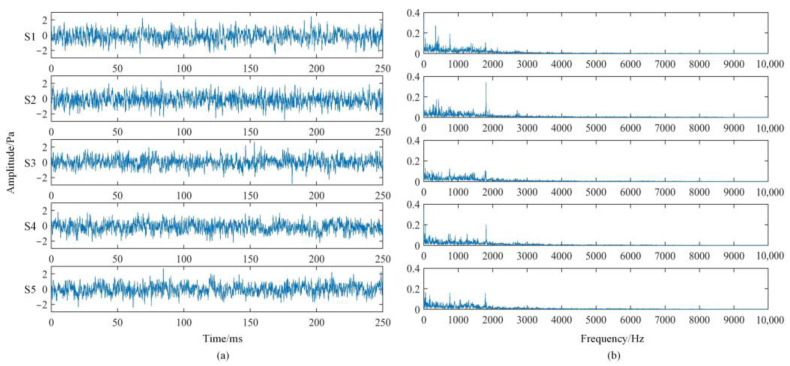
Sound pressure signal in different states: (**a**) time domain diagram; (**b**) frequency domain diagram.

**Figure 8 sensors-22-08325-f008:**
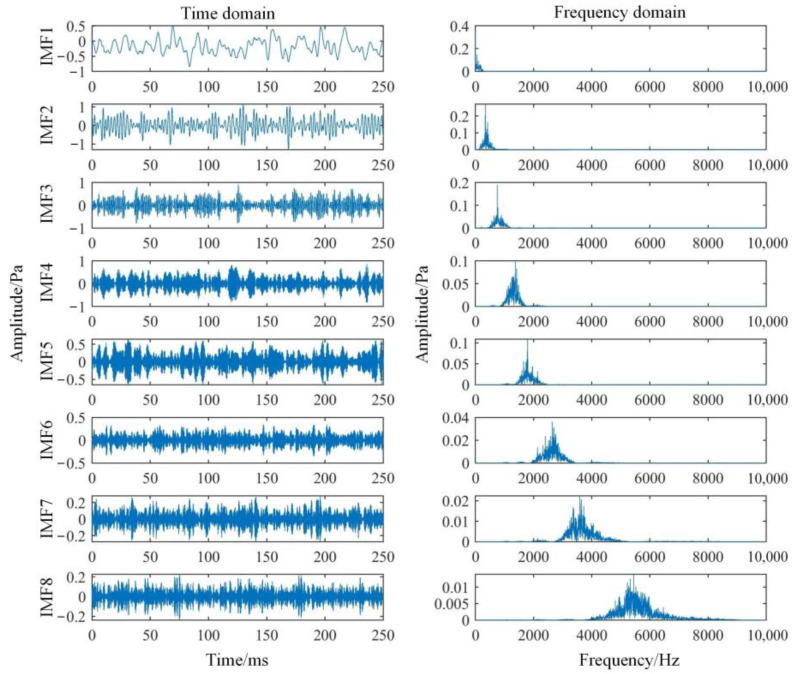
Time and frequency domain waveforms of IMF components.

**Figure 9 sensors-22-08325-f009:**
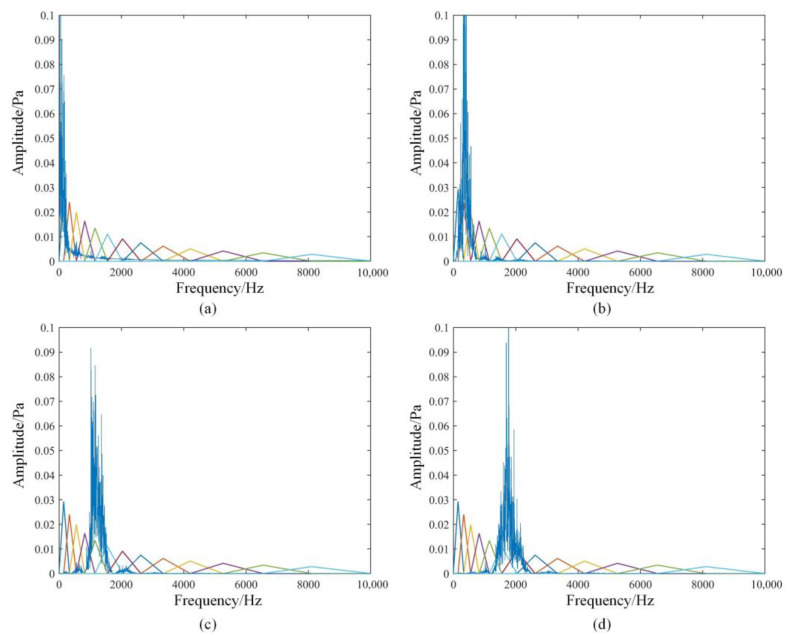
Mapping relationship between IMF and Mel filter: (**a**) IMF_1_; (**b**) IMF_2_; (**c**) IMF_4_; (**d**) IMF_5_.

**Figure 10 sensors-22-08325-f010:**
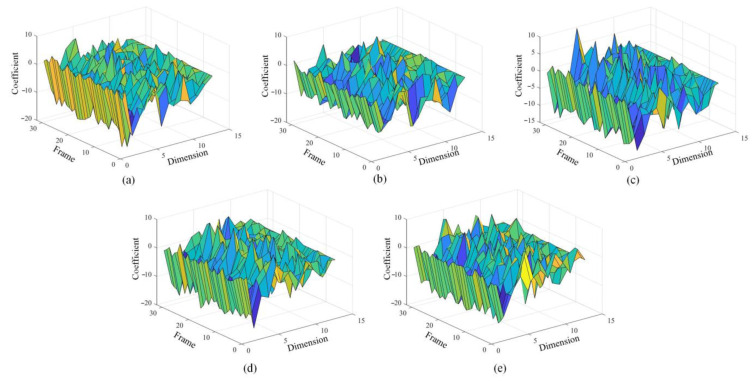
Comparison of MFCC characteristics in different states (**a**) normal state; (**b**) blocked air inlet; (**c**) the first cylinder misfire; (**d**) the third cylinder misfire; (**e**) the sixth cylinder misfire.

**Figure 11 sensors-22-08325-f011:**
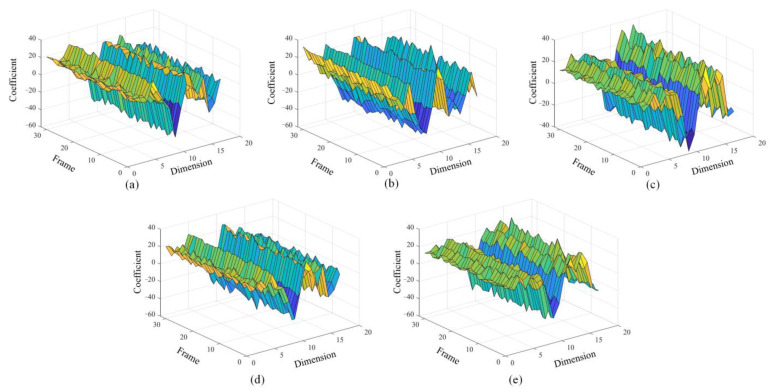
Comparison of VMMFCC characteristics in different states (**a**) normal state; (**b**) blocked air inlet; (**c**) the first cylinder misfire; (**d**) the third cylinder misfire; (**e**) the sixth cylinder misfire.

**Figure 12 sensors-22-08325-f012:**
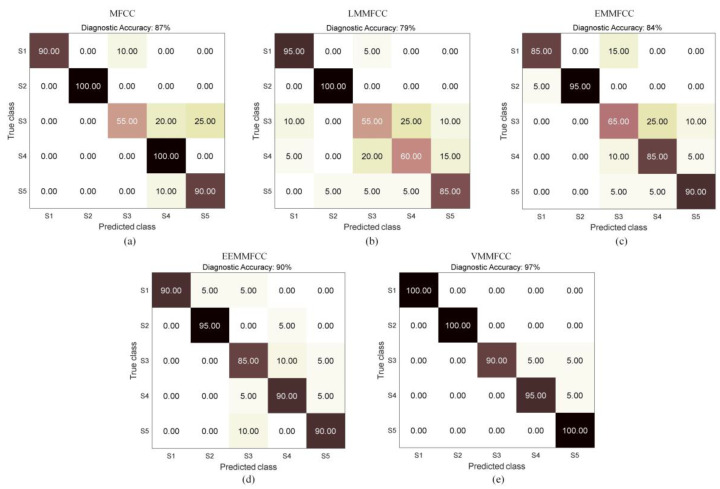
Confusion matrix of fault diagnosis with different features (**a**) MFCC; (**b**) LMMFCC; (**c**) EMMFCC; (**d**) EEMMFCC; (**e**) VMMFCC.

**Figure 13 sensors-22-08325-f013:**
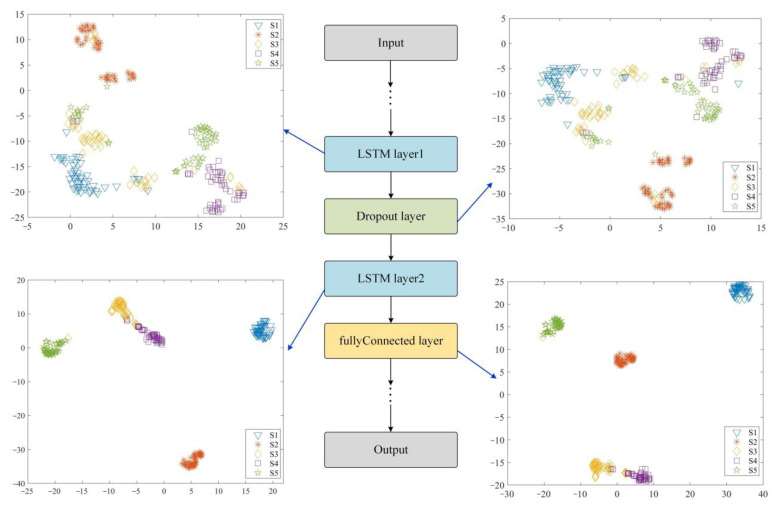
The t−SNE visualization analysis of the VMMFCC−LSTM network for feature extraction.

**Figure 14 sensors-22-08325-f014:**
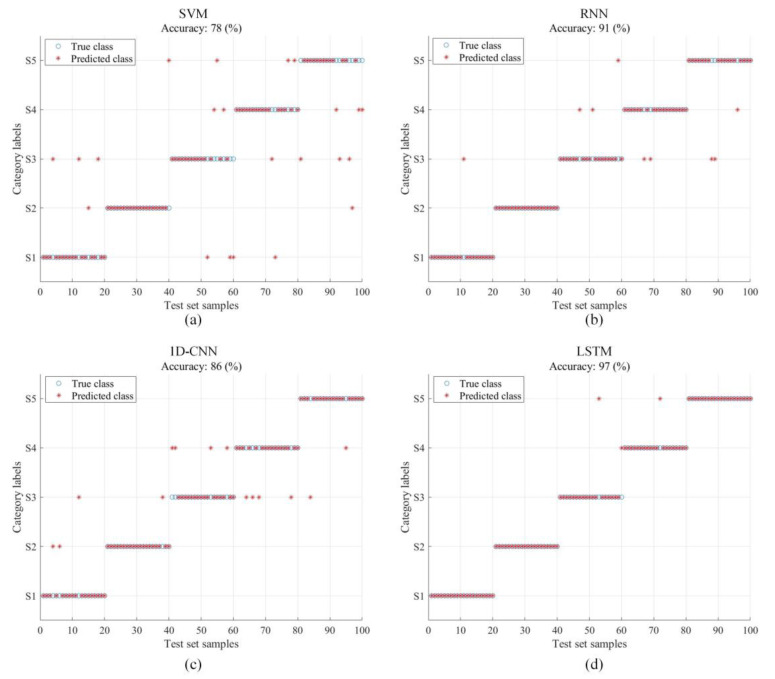
Diagnostic results of VMMFCC in different classifiers (**a**) SVM; (**b**) RNN; (**c**) 1D-CNN; (**d**) EEMMFCC.

**Table 1 sensors-22-08325-t001:** Sampling settings for diesel engine preset states.

No.	Running Status	Rotating Speed	Sampling Frequency	Sampling Time	Number of Sensors
S1	normal status	500 rpm	20 kHz	60 s	2
S2	blocked air inlet	500 rpm	20 kHz	60 s	2
S3	the first cylinder misfire	500 rpm	20 kHz	60 s	2
S4	the third cylinder misfire	500 rpm	20 kHz	60 s	2
S5	the sixth cylinder misfire	500 rpm	20 kHz	60 s	2

**Table 2 sensors-22-08325-t002:** The division of data set sample.

No.	Total Number of Samples	Number of Training Samples	Number of Validation Samples
S1	120	100	20
S2	120	100	20
S3	120	100	20
S4	120	100	20
S5	120	100	20
Total	600	500	100

**Table 3 sensors-22-08325-t003:** Spectral centroid values of IMF components under different decomposition levels *K*.

IMF	Decomposition Levels *K*							
	2	3	4	5	6	7	8	9
IMF_1_	331.0723	207.5153	142.7331	71.697	57.99569	56.33168	56.0017	55.15293
IMF_2_	1573.246	1007.623	744.3112	446.7148	380.4228	374.7676	373.8702	370.7743
IMF_3_	0	1909.521	1433.890	1047.140	826.8442	801.9578	798.8166	783.0283
IMF_4_	0	0	2770.871	1757.248	1231.698	1191.835	1186.98	1153.387
IMF_5_	0	0	0	2824.727	1797.587	1775.476	1770.775	1724.641
IMF_6_	0	0	0	0	2844.753	2665.919	2637.471	2112.904
IMF_7_	0	0	0	0	0	3872.804	3629.005	2732.561
IMF_8_	0	0	0	0	0	0	5576.862	3679.650
IMF_9_	0	0	0	0	0	0	0	5633.213

**Table 4 sensors-22-08325-t004:** Boundary values of the spectral centroid of the IMF for each state at different K-values.

Value of Spectral Centroid	Status	Decomposition Levels *K*							
		2	3	4	5	6	7	8	9
Minimum	S1	331.07	207.52	142.73	71.69	57.99	56.33	56.00	55.15
	S2	417.30	228.92	187.93	173.81	173.25	171.71	171.85	168.99
	S3	168.34	106.11	80.09	39.18	29.91	21.95	21.64	19.42
	S4	322.32	243.90	196.28	171.97	160.78	158.71	106.02	105.59
	S5	271.20	177.30	159.41	124.57	116.65	59.51	60.33	59.04
Maximum	S1	1573.24	1909.52	2770.87	2824.72	2844.75	3872.80	5576.86	5633.21
	S2	1717.91	1845.23	1886.74	2952.63	6372.27	5787.47	8087.25	8109.01
	S3	1472.63	1976.81	2937.07	2795.13	3856.66	4420.12	5761.55	5844.35
	S4	1590.73	2863.08	4022.57	5658.60	4703.65	6509.39	6679.96	6731.02
	S5	1522.05	1993.73	3328.57	4247.02	4462.41	5511.18	7033.35	7154.32

**Table 5 sensors-22-08325-t005:** Indexes related to the energy proportion of each IMF component.

Indicators	IMF_1_	IMF_2_	IMF_3_	IMF_4_	IMF_5_	IMF_6_	IMF_7_	IMF_8_
APE (%)	25.57	21	17.75	15.44	12.22	4.81	1.97	1.23
VPE	0.00252	0.00119	0.00057	0.00066	0.00193	0.00109	0.00010	0.00001
APCC	0.5267	0.5333	0.4932	0.4695	0.3794	0.2472	0.16077	0.1027

**Table 6 sensors-22-08325-t006:** Mapping relationship between each IMF component and Mel filter.

Indicators	IMF_1_	IMF_2_	IMF_4_	IMF_5_	Total
The serial number of the mapping filter	1~4	1~4	3~7	5~8	-
Number of mapping filters	4	4	5	4	17

**Table 7 sensors-22-08325-t007:** Diagnostic accuracy of different characteristics.

Features	Classification Model	Diagnostic Accuracy					Overall Diagnostic Accuracy	Time of Feature Extraction
		S1	S2	S3	S4	S5		
MFCC	LSTM	90%	100%	55%	100%	90%	87%	1.401 s
LMFCC	LSTM	95%	100%	55%	60%	85%	79%	2.337 s
EMFCC	LSTM	85%	95%	65%	85%	90%	84%	2.641 s
EEMFCC	LSTM	90%	95%	85%	90%	90%	90%	4.523 s
VMMFCC	LSTM	100%	100%	90%	95%	100%	97%	3.012 s

**Table 8 sensors-22-08325-t008:** The name of network layer and parameter settings.

No.	Name of Network Layer	Type of Network Layer	Setting of Relevant Parameters	Input Size	Output Size	Number of Learning Parameters
1	Input	Sequenceinput layer	Input dimension: 17	17 × 32	17 × 32	-
2	LSTM1	LSTM layer	Number of hidden cells: 100	17 × 32	100 × 32	4 × [100 × (100 + 17) + 100] = 47,200
3	Dropout	Dropout layer	Dropout rate: 20%	100 × 32	100 × 32	-
4	LSTM2	LSTM layer	Number of hidden cells: 100	100 × 32	100 × 1	4 × [100 × (100 + 100) + 100] = 80,400
5	FC	fullyConnected layer	-	100 × 1	5 × 1	5 × (100 + 1) = 505
6	Softmax	Softmax layer	-	5 × 1	5 × 1	-
7	Output	Classification layer	-	5 × 1	5 × 1	-

**Table 9 sensors-22-08325-t009:** Comparison of different parameter Settings in LSTM network.

Number of Hidden Cells of LSTM	Dropout Rate	The Sum of Trainable Parameters	Overall Diagnostic Accuracy
100	20%	128,105	97%
	10%	128,105	95%
	30%	128,105	96%
50	10%	34,455	95%
	20%	34,455	94%
	30%	34,455	91%
80	10%	83,445	93%
	20%	83,445	95%
	30%	83,445	93%
120	10%	182,365	93%
	20%	182,365	96%
	30%	182,365	95%

**Table 10 sensors-22-08325-t010:** Accuracy of different diagnostic models.

Features	Classification Model	The Sum of Trainable Parameters	Diagnostic Accuracy					Overall Diagnostic Accuracy
			S1	S2	S3	S4	S5	
VMMFCC	LSTM	128,105	100%	100%	90%	95%	100%	97%
VMMFCC	SVM	27,700	80%	95%	70%	80%	65%	78%
VMMFCC	RNN	130,015	95%	100%	85%	90%	85%	91%
VMMFCC	1D-CNN	182,729	85%	95%	80%	80%	90%	86%

## Data Availability

Not applicable.

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
