# Peer review of "Combination of VMD Mapping MFCC and LSTM: A New Acoustic Fault Diagnosis Method of Diesel Engine"

_sensors, 2022, doi:10.3390/s22218325_

Round 1
Reviewer 1 Report
The author has presented a diesel engine fault diagnosis method based on VMD mapping MFCC and LSTM. The VMD method mainly realizes reliable decomposition and noise reduction of the sound pressure signals. The MFCC method achieves effective dimension reduction and feature extraction. LSTM is applied as an advanced diagnostic classification method. The advantages of the above three methods are combined to achieve a successful diagnosis of valve and fire faults in diesel engines. The work of the article is worth affirming, but the following questions remain to be explained.
1. The K-parameter of VMD is set to 8 and the optimization conditions are defined. However, which kind of fault status signal determines this parameter does not seem to be given. That is, whether the fault and normal signal K are the same and whether the optimization conditions of the two types of signals are the same.
2. In Table 4, the author calculates each component's energy magnitude and variation, but only gives very subjective selection conditions. As with Question 1, this may confuse the selection of fault and normal signals. Please propose objective selection conditions
3. The sum of trainable parameters in deep networks is often used as one of the indexes to judge the network scale and has a great influence on the training results. As shown in Table 7, the network parameters are given. However, the deep-learning methods in Table 8 lack this indicator. Please add this indicator to ensure the fairness of training among multiple deep networks.
Author Response
We really appreciate you for your valuable questions and suggestions, which make the logic of this article more reasonable.Now we have completed the modification.Please see the attachment.

Reviewer 2 Report
1.All the abbreviation used need to be clearly defined especially in abstract ( for example VD, MFCC,LSTM ..etc)
2. the validation in page 11 (table 1) need to be improved. a. does 60s measurement adequate? pls justify
b. the speed chosen at 500 rpm for diesel engine is well below idle speed, this is not normal condition for engine

Author Response

(The authors gave the same response as above.)

Round 2
Reviewer 1 Report
The author revised the article according to the suggestions, but there are some problems in the article that need to be corrected:
1. Check the equation, such as Eq. (4)..
2. There is a formatting error in the cross-referencing "Error! Reference source not found".
3. The picture used is relatively fuzzy, improve the resolution of the picture.
Author Response
We really appreciate you for your review. According to your suggestions, we have finished the modification. Please see the attachment.
